METHODS AND RESOURCES

# New approaches to meta-analyze differences in skewness, kurtosis, and correlation

Pietro Pollo[1,2*], Szymon M. Drobniak[1,3], Hamed Haselimashhadi[4º],
Malgorzata Lagisz[1,5º], Ayumi Mizuno[5º], Laura A. B. Wilson[6,7,8º], Daniel W. A. Noble[9‡],
Shinichi Nakagawa[1,5*‡]

1 Evolution & Ecology Research Centre, School of Biological, Earth & Environmental Sciences, University of New South Wales, Sydney, Australia, 2 School of Environmental and Life Sciences, University of Newcastle, Newcastle, Australia, 3 Institute of Environmental Sciences, Faculty of Biology, Jagiellonian University, Kraków, Poland, 4 European Bioinformatics Institute, European Molecular Biology Laboratory, Hinxton, United Kingdom, 5 Department of Biological Sciences, University of Alberta, Biological Sciences Building, Edmonton, Canada, 6 School of Archaeology and Anthropology, The Australian National University, Acton, Australia, 7 School of Biological, Earth and Environmental Sciences, University of New South Wales, Kensington, Australia, 8 ARC Training Centre for Multiscale 3D Imaging, Modelling and Manufacturing, Research School of Physics, The Australian National University, Acton, Australia, 9 Division of Ecology and Evolution, Research School of Biology, The Australian National University, Canberra, Australia

☉ These authors contributed equally and are listed alphabetically.
‡ These authors share senior authorship on this work.
* pietro_pollo@hotmail.com (PP); snakagaw@ualberta.ca (SN)

## Abstract

Biological differences between males and females are pervasive. Researchers often focus on sex differences in the mean or, occasionally, in variation, albeit other measures can be useful for biomedical and biological research. For instance, differences in skewness (asymmetry of a distribution), kurtosis (heaviness of a distribution's tails), and correlation (relationship between two variables) might be crucial to improve medical diagnosis and to understand natural processes. Yet, there are currently no meta-analytic ways to measure differences in these metrics between two groups. We propose three effect size statistics to fill this gap: Δsk, Δku, and ΔZr, which measure differences in skewness, kurtosis, and correlation, respectively. Besides presenting the rationale for the calculation of these effect size statistics, we conducted a simulation to explore their properties and used a large dataset of mice traits to illustrate their potential. For example, in our case study, we found that females show, on average, a greater correlation between fat mass and heart weight than males. Although calculating Δsk, Δku, and ΔZr will require large sample sizes of individual data, technological advancements in data collection create increased opportunities to use these effect size statistics. Importantly, Δsk, Δku, and ΔZr can be used to compare any two groups, allowing a new generation of meta-analyses that explore such differences and potentially leading to new insights in multiple fields of study.

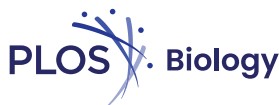

**Data availability statement:** All data and code used in this study are available at https://github.com/pietropollo/new_effect_size_statistics and https://zenodo.org/records/18386956.

**Funding:** SMD was supported by a National Science Centre (Poland; https://www.ncn.gov.pl/) grant (UMO-2020/39/B/NZ8/01274). ML was supported by an ARC (Australian Research Council; https://www.arc.gov.au/) Discovery Project grant (DP230101248). LABW was supported by an ARC Future Fellowship grant (FT200100822). DWAN was supported by an ARC Future Fellowship (FT220100276). SN was supported by an ARC Discovery Project grant (DP210100812) and the Canada Excellence Research Chair Program (CERC-2022-00074; https://www.cerc.gc.ca/). The funders had no role in study design, data collection and analysis, decision to publish, or preparation of the manuscript.

**Competing interests:** The authors have declared that no competing interests exist.

**Abbreviations:** IMPC, International Mouse Phenotyping Consortium; ku, kurtosis; sk, skewness.

## Background

Sex is a biological attribute that can strongly impact organisms' traits, with differences between males and females being central to questions in the biological sciences [1,2]. In contrast, biomedical research has primarily focused on male subjects [3], posing a danger to female health [4,5]. Aware of these issues, the US National Institutes of Health and other health agencies have demanded using multiple sexes in animal studies when possible [6]. As a consequence, the number of biological and biomedical studies using both female and male animals as research subjects has increased in the last decade [7], leading to the accumulation of data that can be used to synthesize and quantify sex differences across biological domains.

Realizing the accumulation of sex-specific data, many perspective pieces have encouraged researchers to investigate sex differences more carefully [8–10]. Yet, some of these pieces, and most of the biological literature, focus exclusively on mean differences between males and females. A fixation on mean differences has been present for a long time in science because researchers tend to focus on dimorphism in trait averages [11], lack sufficiently powerful data, or have limited statistical tools available (or difficulty using them). Yet, measures such as variance, correlation, skewness, and kurtosis can be critical to understanding sex differences. For example, certain traits in mice may exhibit no disparity in average values between sexes, but substantial differences emerge in terms of variability [12,13]. These differences could be more easily assessed because of an effect size statistic that measures differences in variability between two groups (proposed by [14]), illustrating how novel statistical tools can expand possible research questions and provide new scientific insights, such as identifying sex differences in trait selection or canalization.

Beyond variability, the relative shape of trait distributions to the normal distribution (measured by skewness and kurtosis, i.e., asymmetry of a distribution and heaviness of a distribution's tails, respectively; Fig 1A and 1B) can also be crucial to understanding ecological and evolutionary processes and patterns [15–19], as well as improving medical diagnostics [20,21]. For instance, skewness can bias heritability estimates because evolutionary biologists assume that phenotypic components (genetic and environmental) are normally distributed [18]. Furthermore, kurtosis can be used to understand community assembly processes [16]. Besides the shape of trait distributions, evolutionary biologists and quantitative geneticists can quantify correlation matrices to understand trait plasticity and evolvability [22–24], which could then be used for group comparisons (as in [25]; Fig 1C). Although location-scale-shape models [26–28] may be used to explore between-group differences (e.g., males and females) in skewness, kurtosis, or within-group correlations, there are no effect size statistics that can easily measure such differences (but see also [29]).

Here, we propose three new effect size statistics to evaluate between-group differences in skewness ($\Delta sk$), kurtosis ($\Delta ku$), and correlation ($\Delta Zr$), key moments of a distribution that are usually unexplored. These effect size statistics will be valuable to explore sex differences but can also be applied in other fields of study and used to compare differences between any two groups of interest. Meta-analyses using these new effect sizes will create multiple avenues for novel biological enquiries.

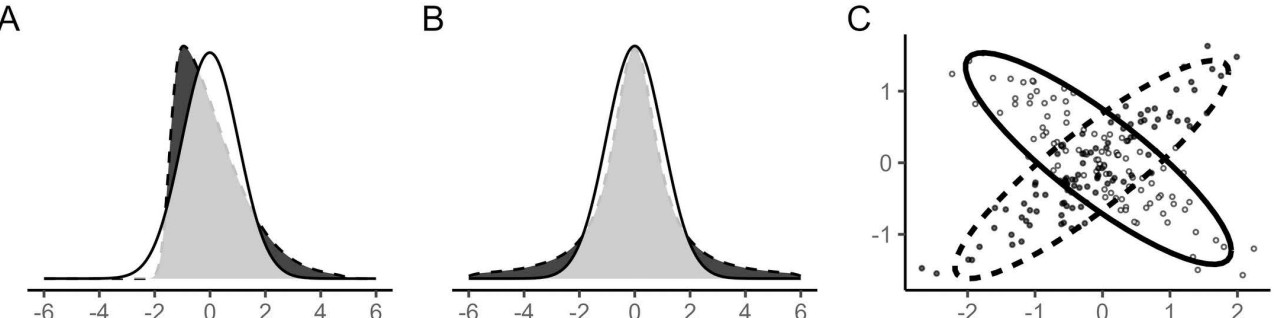

**Fig 1. Simulated trait distributions for two groups with different shapes (A: distinct skewness, B: distinct kurtosis), and different correlations between two traits for two groups (C).** The data and code needed to generate this Figure can be found in https://zenodo.org/records/18386956.

The present moment is particularly conducive for analyses using these new effect sizes because the individual-level data (e.g., individual participant data [30,31]) required for their calculation are increasingly available from new technological advances that allow faster data collection and sharing (e.g., automated phenotyping).

## Difference in skewness and kurtosis

The mean and variance represent the first and second moments of a distribution, respectively. However, the third and fourth moments of a distribution (i.e., skewness and kurtosis, respectively) can also be valuable as they characterize the distribution's shape. More specifically, skewness reflects the distribution's asymmetry around its mean. While positive skewness indicates an elongated right tail with an excess of high values, negative skewness suggests an elongated left tail with an excess of low values. This asymmetry can influence the interpretation of means and variation, as the mean tends to be larger than the median in positively skewed distributions, while the mean tends to be smaller than the median in negatively skewed distributions. Note that a perfectly normal distribution is symmetric (i.e., skewness = 0), where the mean is equal to the median. Sample skewness ($sk$) [32] can be expressed as:

$$sk = \frac{\frac{1}{n}\sum_{i=1}^{n}(x_i - \bar{x})^3}{\left[\frac{1}{n}\sum_{i=1}^{n}(x - \bar{x})^2\right]^{\frac{3}{2}}} \frac{\sqrt{n(n-1)}}{n-2}$$

(1)

where $x_i$ is a raw data value, $\bar{x}$ is the sample mean, and $n$ is the sample size. Skewness sampling variance ($s^2_{sk}$) [32] can then be expressed as:

$$s^2_{sk} = \frac{6n(n-1)}{(n-2)(n+1)(n+3)}$$

(2)

On the other hand, kurtosis measures tail heaviness: high kurtosis distributions have heavier tails (i.e., proportionally more extreme values than central values), whereas low kurtosis distributions have lighter tails. For comparison, a normal distribution is expected to have kurtosis = 3. Sample excess kurtosis ($ku$) [32] can be expressed as:

$$ku = \frac{n(n+1)(n-1)}{(n-2)(n-3)} \frac{\sum_{i=1}^{n}(x_i - \bar{x})^4}{\left[\sum_{i=1}^{n}(x_i - \bar{x})^2\right]^2} - \frac{3(n-1)^2}{(n-2)(n-3)}$$

(3)

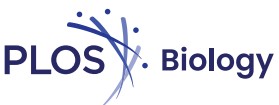

with sampling variance ($s^2_{ku}$) [32] as:

$$s^2_{ku} = \frac{24n(n-1)^2}{(n-3)(n-2)(n+3)(n+5)}$$
(4)

Evaluating skewness and kurtosis provides valuable insights into a variable distribution, which is crucial for interpreting means, assessing variability, and making informed decisions in statistical analyses. Although meta-analyses can use skewness (Eq 1) and kurtosis (Eq 3) to investigate single variables, effect size statistics that compare these metrics between two groups are lacking. Thus, we propose the difference between two groups in skewness ($\Delta sk$), expressed as:

$$\Delta sk = sk_1 - sk_2$$
(5)

and its sampling variance ($s^2_{\Delta sk}$) as:

$$s^2_{\Delta sk} = s^2_{sk_1} + s^2_{sk_2} - 2\rho_{sk}s_{sk_1}s_{sk_2}$$
(6)

where $\rho_{sk}$ represents the sampling correlation in skewness between the two groups (zero if assumed to be independent). Similarly, we propose the difference between two groups in kurtosis ($\Delta ku$), expressed as:

$$\Delta ku = ku_1 - ku_2$$
(7)

and its sampling variance ($s^2_{\Delta ku}$) as:

$$s^2_{\Delta ku} = s^2_{ku_1} + s^2_{ku_2} - 2\rho_{ku}s_{ku_1}s_{ku_2}$$
(8)

where $\rho_{ku}$ represents the sampling correlation in kurtosis between the two groups (zero if assumed to be independent).

However, we note that Eqs 2 and 4 assume normality for sampling variances. When the underlying distributions are skewed or heavy-tailed, sampling error variances for skewness and kurtosis (Eqs 2 and 4) and, by extension, for their between-group contrasts (Eqs 5–8), can misestimate uncertainty. To assess robustness and to provide distribution-free alternatives, we complemented the analytic formulas with resampling-based estimators computed within each group and summed for the difference (i.e., jackknife [33]; see our simulation study below).

## Difference in correlation

Numerous meta-analyses estimate the correlation between two variables [34,35]. To do so, researchers use the effect size statistic $Zr$ [36], which can be expressed as:

$$Zr = \frac{ln\left(\frac{1+r}{1-r}\right)}{2}$$
(9)

and its sampling variance ($s^2_{Zr}$) [36] as:

$$s^2_{Zr} = \frac{1}{n-3}$$
(10)

where $r$ is Pearson's correlation coefficient between two variables and $n$ is the sample size.

Although $Zr$ alone remains extremely useful to test correlational hypotheses, researchers from all fields would benefit from being able to compare $Zr$ values between two groups. Although Cohen [37] proposed the difference between two



groups in $Zr$ as $q$, he did not provide an equation to calculate its sampling variance. Consequently, this effect size statistic has not been used despite its potential. We therefore propose the difference between two groups in $Zr$ with a new name ($\Delta Zr$), as:

$$\Delta Zr = Zr_1 - Zr_2 \tag{11}$$

and its sampling variance ($s^2_{\Delta Zr}$) as:

$$s^2_{\Delta Zr} = s^2_{Zr_1} + s^2_{Zr_2} - 2\rho_{Zr}s_{Zr_1}s_{Zr_2} \tag{12}$$

where $\rho_{Zr}$ represents the sampling correlation in Fisher's $Zr$ between the two groups (zero if assumed to be independent).

## Simulation study

We conducted Monte-Carlo simulations to evaluate bias and variance estimation for our new effect sizes $\Delta sk$, $\Delta ku$, and $\Delta Zr$. For $\Delta sk$ and $\Delta ku$, we simulated independent samples for two groups from Pearson distributions with known moments using the *rpearson* function from the R package *PearsonDS* v. 1.3.2 [38]. We conducted two simulations: (1) by changing skewness between groups that involved moderate departures from normality in which group-specific skewness from $sk$ ∈ {−1, −0.5, 0, 0.5, 1} and kurtosis was fixed at 3; (2) by holding skewness constant ($sk = 0$) while manipulating kurtosis from $ku$ ∈ {2.5, 3, 4, 5, 6}. In all cases, we simulated scenarios where: (i) the variance between each group was the same ($\sigma^2_2 = \sigma^2_1 = 1$) or different ($2\sigma^2_2$ versus $\sigma^2_1$); (ii) the mean between the two groups was the same ($u_2 = u_1 = 0$) or different ($u_2 = 5$, $u_1 = 0$). For simplicity, we assumed equal sample sizes between groups with sample size varying from $n$ ∈ {10, 20, …, 100, 150, 500}. We created all unique combinations of the above scenarios resulting in 1,200 independent scenarios (when considering each of the 100 scenarios at each sample size). We estimated $\Delta sk$ and $\Delta ku$ for each scenario using formulas for within-group sample skewness with small-sample correction (Eq 1) and excess kurtosis with small-sample correction (Eq 3) to estimate point estimates. To estimate associated sampling variance for $\Delta sk$ and $\Delta ku$ we used the analytical variance estimators derived here (Eqs 2 and 4) and an associated re-sampling (jackknife) approach to compute group sampling variances separately followed by pooling. Importantly, our simulations assume no correlation between groups.

For $\Delta Zr$ simulations, we simulated two groups each containing two variables with known correlations within each group. For $\Delta Zr$, we drew bivariate normal data with target within-group correlations r ∈ {−0.8, −0.4, −0.2, 0, 0.2, 0.4, 0.6, 0.8} using the *mvnorm* function from the package *MASS* v. 7.3.61 [39]. Marginals were standard normal and group sizes varied from n ∈ {10, 20, …, 100, 150, 500}. We created all unique combinations of scenarios resulting in 768 unique scenarios. We estimated $\Delta Zr$ using Fisher's Z transformation $Zr$ and calculating $\Delta Zr$ as the difference of $Zr$ across groups (Eqs 9–11). Sampling variance for $\Delta Zr$ used Eq 10 and a jackknife approach. Again, we assumed no correlation between our groups.

Note that our simulations did not explore differences in sample size between groups. However, many groups being compared in meta-analyses have the same or very similar sample size. Additionally, simulations often show relatively small impacts of unbalanced sample sizes [40,41], which is why we originally did not vary sample size between groups in our simulations.

We resampled 2,500 times for each scenario across all simulations. Performance metrics were (a) bias of the point estimator, (b) relative bias of the sampling-variance estimator, (c) coverage (95%), and (d) Monte-Carlo standard errors (MCSEs). See Supporting information for full formulas. We also evaluated the performance of these effects for meta-analysis (see details in Sections 8.4 and 9.4 of the Supporting information).

## Simulation results

In all cases, we found the Monte Carlo standard error (MCSEs) to be low for all our performance metrics (range of MCSEs for $\Delta sk$: 0 to 0.01; $\Delta ku$: 0 to 0.624; $\Delta Zr$: 0 to 0.004). $\Delta sk$, $\Delta ku$, and $\Delta Zr$ point estimators exhibited small sample bias with

less than 20–30 samples, except for Δ*ku*, which showed this bias below *n* < 50–60, indicating effect sizes involving kurtosis are more challenging to estimate (S1 and S2 Figs). Differences in the mean and variance between groups did not differentially affect bias (S3 Fig). Regardless, small sample biases were moderate, and there was rarely a consistent over- or under-estimation in point estimates across the scenarios evaluated (S1 Fig). Bias-corrected jackknife estimates reduced the small-sample bias relative to analytical bias corrected-moment estimators (mean square bias, jackknife, and analytical, for Δ*sk*: 1.109, 3.375; Δ*ku* 477.71, 891.659; Δ*Zr* 0.029, 0.214).

In contrast to point estimators, the effectiveness of sampling variance estimators for Δ*sk*, Δ*ku*, and Δ*Zr* varied. Analytical sampling variance formulas for Δ*sk* and Δ*ku* were consistently biased (S4 Fig). Jackknife resampling when combined with analytical point estimates (Fig 2) performed the best. Under these conditions, estimators performed well when *n* > 50. In contrast, the performance of sampling variance estimators for Δ*Zr* was best when using the analytical formulas for both the point estimator and its associated sampling variance (Fig 2).

Coverage was close to nominal (95%) for Δ*sk* and Δ*Zr* across sample sizes (Fig 2C and 2I). Coverage for Δ*ku*, however, was poor across many simulated scenarios (Fig 2F). Increased sample size did not improve coverage. Poor coverage was the result of skewed sampling distributions from Jackknife approaches (S5 and S6 Figs). At small sample sizes, Δ*ku* was estimated poorly when true Δ*ku* was high, leading to non-skewed distributions with good coverage. In contrast,

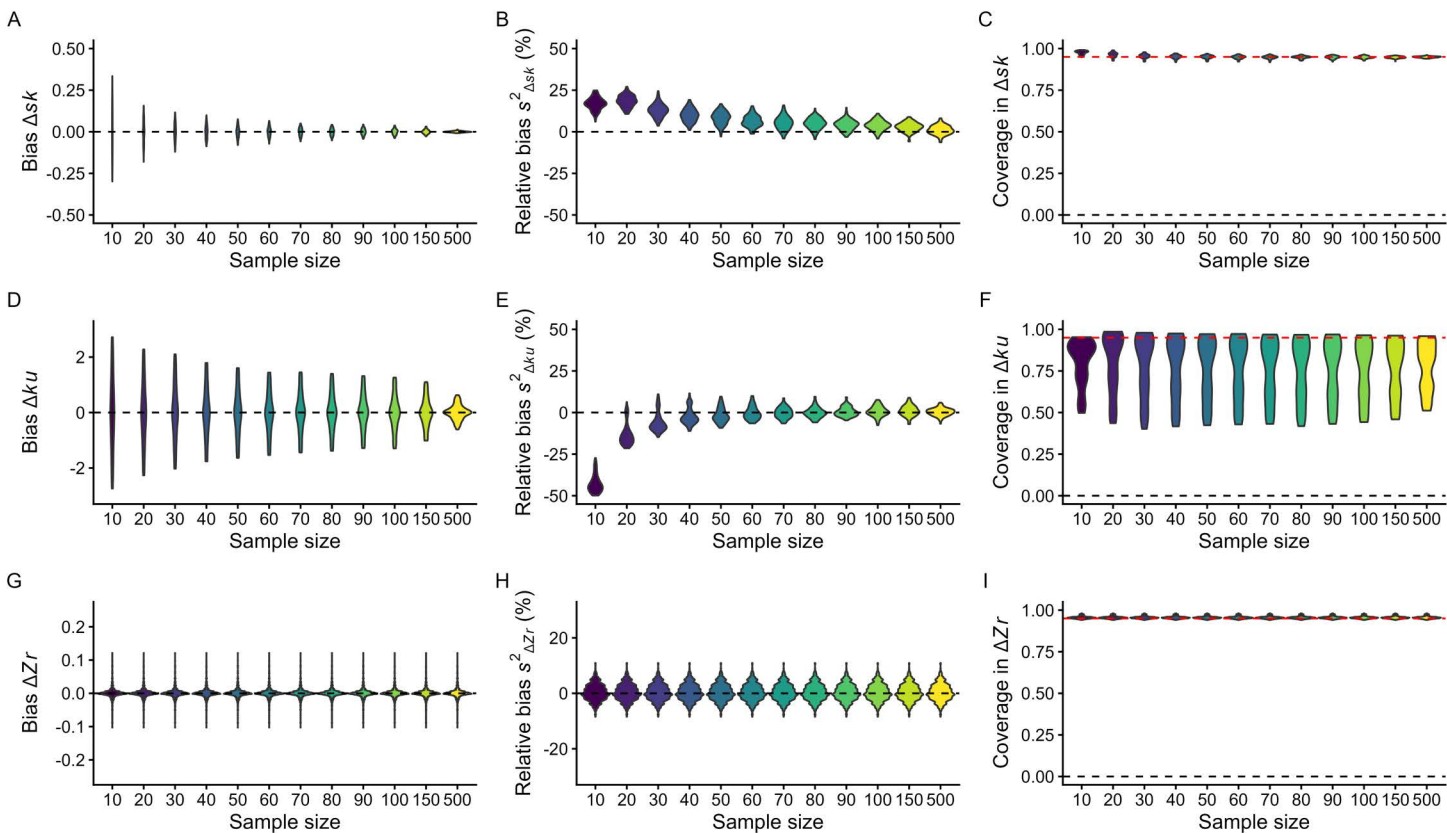

**Fig 2. Bias in Δ*sk*, Δ*ku*, and Δ*Zr* effect estimates (A, D, G), relative bias in their sampling variance using jackknife-based approximation (B, E, H), and coverage of effect estimates (C, F, I) across simulations where samples ranged in group sample sizes between *n* ∈ {10, 20, …, 100, 150, 500}.** A total of 100 simulated scenarios were assessed for Δ*sk* and Δ*ku* whereas 64 simulated scenarios were assessed for Δ*Zr*. We ran 2,500 simulations for each scenario. For simplicity, we only present results from our recommended point estimators and sampling variance estimators using jackknife. See supplementary material for full simulation results. The data and code needed to generate this Figure can be found in https://zenodo.org/records/18386956.

large sample sizes improved point estimation of Δ*ku* when differences existed, but the sampling distribution became highly skewed leading to poor coverage (S5 and S6 Figs). These problems stem from the fact that the standard error formula for kurtosis assumes normality (see [42]).

Considering these simulation results, we suggest pairing the formula-based point estimators for skewness (Eq 1) and kurtosis (Eq 3) with jackknife standard errors for Δ*sk* and Δ*ku*. For Δ*Zr*, the standard analytic variance is recommended (Eqs 9–12). This choice balances efficiency under normality with robustness to realistic deviations from it and aligns with our broader guidance to avoid very small group sizes for these statistics. Given the challenges in estimating Δ*ku,* and the poor properties of its sampling variance [42], we recommend weighted meta-analytic models using sample size instead of sampling variance (see Supporting information and [41]).

## Worked examples: Sex differences in mice

To illustrate the application of our proposed effect size statistics, we used data compiled by the International Mouse Phenotyping Consortium (IMPC, version 18.0; [43]; http://www.mousephenotype.org/). We examined differences between male and female mice in two pairs of traits from distinct functional domains: morphology (fat mass and heart weight) and physiology (glucose and total cholesterol). We selected these traits because they are widely understood traits, even by non-specialists, and had a large sample size (more than 10,000 individuals measured). More specifically, we assessed differences between the sexes in mean (using the natural logarithm of the response ratio [44], hereby *lnRR*), variability (using the natural logarithm of the variance ratio [14], hereby *lnVR*), skewness (using Δ*sk*), and kurtosis (using Δ*ku*) for each trait, as well as in the difference in correlation for each trait pair (using Δ*Zr*). The IMPC dataset contains data from multiple phenotyping centers and mice strains, so we selected the ones with the most data points for our analyses here, computing the aforementioned effect size statistics separately for each one of them.

We performed a meta-analysis for each effect size statistic to obtain a mean effect size for each trait (or pair of traits, in the case of Δ*Zr*), using "effect size ID," "phenotyping center," and "mice strain" as random factors in meta-analytical models (due to substantial heterogeneity, Table 1). In the case of Δ*ku,* we fitted a weighted meta-analytic model using sample size instead of sampling variance (see previous sections and [41]). In all these analyses, positive effect sizes denoted a greater estimate (mean, variability, skewness, kurtosis, or correlation) for males than females. We conducted all statistical analyses in the software R 4.5.1 [45]. We used the functions *moment_effects* and *cor_diff*, which have been incorporated into the package *orchaRd* v. 2.1.3 [46], to compute Δ*sk*, Δ*ku*, and Δ*Zr*. We fitted meta-analytical models using the *rma.mv* function from the package *metafor* v. 4.8-0 [47]. All methodological details and additional information can be found in our tutorial, at https://pietropollo.github.io/new_effect_size_statistics/.

We found that males, on average, had greater fat mass and heart weight than females regardless of phenotyping center and mice strain (Fig 3A, 3B, 3F, and 3G). The variability among individuals regarding these traits was also greater for males than for females, except for fat mass from one specific phenotyping center and mice strain (Fig 3C). By contrast, females had a similar skewness in fat mass and heart weight compared with males (Fig 3D and 3I). However, Δ*sk* values for fat mass and heart weight varied across phenotyping centers and mice strains, with negative and positive values present (Fig 3D and 3I). Sex differences in kurtosis for fat mass and heart weight followed a very similar pattern to the one described for skewness: Δ*ku* values overlapping zero with some variation across individual effect sizes (Fig 3E and 3J). Moreover, the correlation between fat mass and heart weight was, on average, greater for females than males (Fig 4A and 4B). However, this difference in correlation was absent for some phenotyping centers and mice strains (Fig 4A and 4B).

We also found that male and female mice were, on average, similar in terms of blood glucose levels (Fig 5A and 5B), although males had higher total cholesterol than females (Fig 5F and 5G). We observed the same pattern regarding the variability of these traits: on average, the sexes were similarly variable in glucose (Fig 5C), but the variability of total cholesterol was greater in males than in females (Fig 5H). Contrasting with morphological traits, sex differences in skewness and kurtosis were mostly absent (Fig 5D, 5E, 5I, and 5J). Lastly, males and females showed a similar relationship



**Table 1. Heterogeneity estimates ($I^2$) for each meta-analytical model fitted in our study.**

| Trait(s) | Effect size type | $I^2_{total}$ | $I^2_{effect\ size\ ID}$ | $I^2_{phenotyping\ center}$ | $I^2_{strain}$ |
|---|---|---|---|---|---|
| Fat mass | *lnRR* | 97.69 | 97.69 | <0.01 | <0.01 |
| Fat mass | *lnVR* | 95.71 | < 0.01 | 33.60 | 62.11 |
| Fat mass | Δ*sk* | 75.61 | 9.82 | <0.01 | 65.80 |
| Fat mass | Δ*ku* | 85.81 | < 0.01 | <0.01 | 85.81 |
| Heart weight | *lnRR* | 96.32 | 69.24 | <0.01 | 27.08 |
| Heart weight | *lnVR* | 87.15 | 87.15 | <0.01 | <0.01 |
| Heart weight | Δ*sk* | 68.48 | 38.29 | 30.19 | <0.01 |
| Heart weight | Δ*ku* | 97.90 | <0.01 | 84.60 | 13.30 |
| Glucose | *lnRR* | 94.76 | 42.67 | <0.01 | 52.09 |
| Glucose | *lnVR* | 70.08 | <0.01 | 70.08 | <0.01 |
| Glucose | Δ*sk* | 11.76 | <0.01 | 11.76 | <0.01 |
| Glucose | Δ*ku* | 3.60 | <0.01 | 1.21 | 2.14 |
| Total cholesterol | *lnRR* | 95.43 | 69.77 | <0.01 | 25.66 |
| Total cholesterol | *lnVR* | 94.70 | 84.86 | <0.01 | 9.84 |
| Total cholesterol | Δ*sk* | < 0.01 | <0.01 | <0.01 | <0.01 |
| Total cholesterol | Δ*ku* | 68.94 | <0.01 | 68.63 | 0.31 |
| Fat mass and heart weight | Δ*Zr* | 64.87 | <0.01 | 64.87 | <0.01 |
| Glucose and total cholesterol | Δ*Zr* | 92.33 | <0.01 | <0.01 | 92.33 |

between glucose and total cholesterol, albeit this relationship was stronger for males than for females in some instances (Fig 4C and 4D).

Our findings that females have, on average, lower (Fig 3B and 3G), less variable (Fig 3C and 3H), but similar skewness (Fig 3D and 3I) and extreme values (kurtosis; Fig 3E and 3I) of fat mass and heart weight compared with males may contribute to sex-related differences in the development of diseases associated with these traits and their biomarkers (e.g., QTc interval length [48]). Moreover, a stronger relationship between fat mass and heart weight in females than in males (Fig 4B) may represent a greater risk of cardiohypertrophy arising from obesity in the former compared with the latter [49]. Meanwhile, absent or less pronounced sex differences in glucose and total cholesterol (Fig 4) may suggest other sources of variation may contribute to sex differences in the symptomology of diseases associated with these measurements (e.g., [50–52]). Characterizing sex differences in biological traits, as we have done here, can provide new perspectives on evolutionary, ecological, and medical patterns, possibly improving healthcare and environmental interventions.

## Limitations

Despite the enormous potential of the effect size statistics we proposed here, they are not free of limitations. For instance, skewness and kurtosis (and therefore the difference in these estimates between two groups; i.e., Δ*sk* and Δ*ku*, respectively) are more likely to become extreme with small sample sizes and with variables with few unique values, either because the variable is discrete or because it is naturally constant (e.g., number of vertebrae in mice). We thus recommend that researchers only compute Δ*sk* and Δ*ku* for continuous variables with a minimum sample size of 50 for each group (as shown in our simulations). Importantly, we found that Δ*ku* variance estimates can be biased in many situations, highlighting that exploring Δ*ku* should be a priority for future work. Because of this issue, meta-analyzing Δ*ku* requires sample size-based weights instead of the standard sampling variance (see supplementary material and [41]). Lastly, although Δ*sk*, Δ*ku*, and Δ*Zr* can be calculated, respectively, from reported skewness, kurtosis, or within-group correlations for different samples, empirical studies rarely report these estimates. Therefore, calculating these effect sizes will probably require raw data, which, fortunately, are now becoming more readily available.

**Fig 3. Examples of morphological sex differences in mice (fat mass, A–E; heart weight, F–J) for various phenotype centers (each with a different color in panels B–E and G–J) and mice strains (each with a different shape in panels B–E and G–J), with the bottom estimate in panels B–E and G–J (turquoise diamond) representing the mean effect size.** A and F show distributions of these traits (scaled by subtracting the mean from each value and then dividing the result by the standard deviation) for males (black with dashed borders) and females (white with solid borders), with the sample size of females and males shown as Nf and Nm, respectively. Panels B–E and G–J show effect sizes (*lnRR*: natural logarithm of the response ratio; *VR*: variance ratio; Δ*sk*: difference in skewness; Δ*ku*: difference in kurtosis), with their respective point estimate and 95% confidence interval stamped. The data and code needed to generate this Figure can be found in https://zenodo.org/records/18386956.



**Fig 4. Relationship between fat mass and heart weight (A, B) and glucose and total cholesterol (C, D) in mice.** Panels A and C show these relationships (with variables scaled by subtracting the mean from each value and then dividing the result by the standard deviation) separately for males (dashed line) and females (solid line), each subpanel representing a different phenotyping center and/or mice strain, with the sample size of females and males shown as Nf and Nm, respectively. Panels B and D then show differences in correlation (ΔZr) between males and females (point estimate and 95% confidence interval stamped), where each color represents a distinct phenotype center and each shape represents a distinct mice strain, with the bottom estimate in each panel (turquoise diamond) representing the mean effect size. Note that panels A and C contain individual data points, which may appear as background shading in cases with large sample sizes. The data and code needed to generate this Figure can be found in https://zenodo.org/records/18386956.



**Fig 5. Examples of physiological sex differences in mice (glucose, A–E; total cholesterol, F–J) for various phenotype centers (each with a different color in panels B–E and G–J) and mice strains (each with a different shape in panels B–E and G–J), with the bottom estimate in panels B–E and G–J (turquoise diamond) representing the mean effect size.** A and F show distributions of these traits (scaled by subtracting the mean from each value and then dividing the result by the standard deviation) for males (black with dashed borders) and females (white with solid borders), with the sample size of females and males shown as Nf and Nm, respectively. Panels B–E and G–J show effect sizes (*lnRR*: natural logarithm of the response ratio; *VR*: variance ratio; Δ*sk*: difference in skewness; Δ*ku*: difference in kurtosis), with their respective point estimate and 95% confidence interval stamped. The data and code needed to generate this Figure can be found in https://zenodo.org/records/18386956.

## Future opportunities

The effect size statistics proposed in the present study can be useful across the life sciences, social sciences, and medicine. This is because skewness and kurtosis, and consequently differences between any two or more groups in these estimates (i.e., Δ*sk* and Δ*ku*), may help researchers to understand epidemiological trends [53], genetic patterns relevant to medical diagnosis [20,21], disruptive selection on quantitative traits [54], body size patterns across individuals [55] and species [56], reproductive patterns [57], regime shifts in ecosystems [58], heritability [18], community assembly processes [16], and possibly many other topics. Meanwhile, comparisons regarding correlations have been used to explore memory processing during sleep [59], physiological patterns in patients with certain medical conditions [60], and selection patterns [22–24], to name a few. Because Δ*Zr* can be used in virtually any comparison between two groups of correlational data, the opportunities for its use are endless. Most importantly, Δ*sk*, Δ*ku*, and Δ*Zr* are unitless measures, so they can be meta-analyzed to uncover patterns between two groups (e.g., males and females). Moreover, the growing availability of raw data and big data approaches, facilitated by technological advances, makes these effect size statistics particularly valuable for modern research.

## Supporting information

**S1 File. An HTML file containing all steps to reproduce simulations and meta-analyses presented in our study.** (HTML)

**S1 Fig. Bias in Δ*sk*, Δ*ku*, and Δ*Zr* effect estimates across simulations where samples ranged in group sample sizes between *n* ∈ {10, 20, …, 100, 150, 500}.** A total of 100 simulated scenarios were assessed for Δ*sk* and Δ*ku* whereas 64 simulated scenarios were assessed for Δ*Zr*. We ran 2,500 simulations for each scenario. The data and code needed to generate this Figure can be found in https://zenodo.org/records/18386956.
(TIF)

**S2 Fig. Bias of analytical point estimators in relation to the absolute difference in skewness and kurtosis between groups. (A)** Skewness and **(B)** kurtosis. Color of points correspond to the sample size and each point is a single simulated scenario. The dotted line is the zero bias line. The data and code needed to generate this Figure can be found in https://zenodo.org/records/18386956.
(TIF)

**S3 Fig. Bias for Δ*sk* and Δ*ku* for simulated scenarios was not related to group means or variances being different.** We ran 2,500 simulations for each scenario. The data and code needed to generate this Figure can be found in https://zenodo.org/records/18386956.
(TIF)

**S4 Fig. Relative bias in Δ*sk*, Δ*ku*, and Δ*Zr* effect estimates across simulations where samples ranged in group sample sizes between *n* ∈ {10, 20, …, 100, 150, 500}.** A total of 100 simulated scenarios were assessed for Δ*sk* and Δ*ku* whereas 64 simulated scenarios were assessed for Δ*Zr*. Note that for relative bias different combinations of point estimates and sampling variance estimates were used in their calculation as indicated in their titles which show the calculation. Notation is as follows ku and sk are the skewness and kurtosis calculated using original formulas. sk_sv and ku_sv are the sampling variance estimates using the original formulas. jack_skew_sv and jack_ku_sv are the sampling variance estimates for skewness and kurtosis using jackknife. jack_skew_bc and jack_ku_bc are the bias-corrected point estimates from the jackknife. We ran 2,500 simulations for each scenario.
(TIF)

**S5 Fig. Coverage of 95% confidence intervals for Δ*sk*, Δ*ku*, and Δ*Zr* effect estimates across simulations where samples ranged in group sample sizes between *n* ∈ {10, 20, …, 100, 150, 500}.** A total of 100 simulated scenarios

were assessed for $\Delta sk$ and $\Delta ku$ whereas 64 simulated scenarios were assessed for $\Delta Zr$. We ran 2,500 simulations for each scenario. The data and code needed to generate this Figure can be found in https://zenodo.org/records/18386956. (TIF)

**S6 Fig. Example sampling distributions of three different scenarios ($\Delta ku = 0$, 1, or 2.5) for $n = 10$ and $n = 500$ samples for each group.** We ran 2,500 simulations for each scenario. The data and code needed to generate this Figure can be found in https://zenodo.org/records/18386956. (TIF)

## Acknowledgments

We thank Yefeng Yang for his contribution in the early stage of this study.

**Declaration of AI use:** The authors declare that they occasionally used GPT-4-turbo (OpenAI) to improve the clarity and readability of this work. After using these tools, the authors reviewed and edited the content as needed and took full responsibility for the content of the publication.

## Author contributions

**Conceptualization:** Pietro Pollo, Shinichi Nakagawa.

**Data curation:** Pietro Pollo.

**Formal analysis:** Pietro Pollo, Szymon M. Drobniak, Daniel W. A. Noble, Shinichi Nakagawa.

**Funding acquisition:** Shinichi Nakagawa.

**Methodology:** Pietro Pollo, Shinichi Nakagawa.

**Project administration:** Pietro Pollo, Shinichi Nakagawa.

**Software:** Pietro Pollo, Daniel W. A. Noble.

**Supervision:** Daniel W. A. Noble, Shinichi Nakagawa.

**Visualization:** Pietro Pollo, Daniel W. A. Noble.

**Writing – original draft:** Pietro Pollo, Shinichi Nakagawa.

**Writing – review & editing:** Pietro Pollo, Szymon M. Drobniak, Hamed Haselimashhadi, Malgorzata Lagisz, Ayumi Mizuno, Laura A. B. Wilson, Daniel W. A. Noble, Shinichi Nakagawa.

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
