## [Editor Report · Decision Letter 0]

17 Apr 2025

Dear Dr Pollo,

Thank you for submitting your manuscript entitled "Beyond sex differences in mean: meta-analysis of differences in skewness, kurtosis, and correlation" for consideration as a Methods and Resources by PLOS Biology.

Your manuscript has now been evaluated by the PLOS Biology editorial staff, as well as by an academic editor with relevant expertise, and I'm writing to let you know that we would like to send your submission out for external peer review.

Once your full submission is complete, your paper will undergo a series of checks in preparation for peer review. After your manuscript has passed the checks it will be sent out for review. To provide the metadata for your submission, please Login to Editorial Manager (https://www.editorialmanager.com/pbiology) within two working days, i.e. by Apr 22 2025 11:59PM.

Kind regards,

Roli Roberts

Roland Roberts, PhD

Senior Editor

PLOS Biology

rroberts@plos.org

---

## [Decision Letter · Decision Letter 1]

25 Jun 2025

Dear Dr Pollo,

Thank you for your patience while your manuscript "Beyond sex differences in mean: meta-analysis of differences in skewness, kurtosis, and correlation" was peer-reviewed at PLOS Biology. Your manuscript has been evaluated by the PLOS Biology editors, an Academic Editor with relevant expertise, and by three independent reviewers.

You'll see that reviewer #1 is broadly positive, but focuses on the larger samples sizes needed to reliably calculate these higher moments, and feels that your statement on this in the limitations is insufficient; instead, s/he wants a much more substantial discussion of this issue, potentially with further analysis, and a justification of your n=50 threshold. Reviewer #2 is significantly more sceptical, noting that you seem to use “off the shelf” calculations in a rather simplistic way (my interpretation) that undermines his confidence in their use for meta-analysis. He suggests that some significant validation is needed before this will be acceptable. Reviewer #3 (in an attachment) is quite positive, but wonders if the need for n=50 will limit its utility for preclinical studies, and (like rev #2) asks about the provenance of the formulae used in the code.

As you will see in the reviewer reports, which can be found at the end of this email, although the reviewers find the work potentially interesting, they have also raised a substantial number of important concerns. Based on their specific comments and following discussion with the Academic Editor, it is clear that a substantial amount of work would be required to meet the criteria for publication in PLOS Biology. However, given our and the reviewer interest in your study, we would be open to inviting a comprehensive revision of the study that thoroughly addresses all the reviewers' comments. Given the extent of revision that would be needed, we cannot make a decision about publication until we have seen the revised manuscript and your response to the reviewers' comments. Your revised manuscript would need to be seen by the reviewers again, but please note that we would not engage them unless their main concerns have been addressed.

Having discussed the reviews with the Academic Editor, we think you should address all of the reviewers' comments, but in particular the requests from reviewer #2 for a more thorough characterisation and validation of your approach.

We appreciate that these requests represent a great deal of extra work, and we are willing to relax our standard revision time to allow you 6 months to revise your study. Please email us (plosbiology@plos.org) if you have any questions or concerns, or envision needing a (short) extension.

**IMPORTANT - SUBMITTING YOUR REVISION**

*Resubmission Checklist*

*Published Peer Review*

*PLOS Data Policy*

*Blot and Gel Data Policy*

Sincerely,

Roli Roberts

Roland Roberts, PhD

Senior Editor

PLOS Biology

rroberts@plos.org

REVIEWERS' COMMENTS:

Reviewer #1:

This is an interesting manuscript proposing to extend meta-analyses by including measures of skewness and kurtosis and correlations of variables. The focus, here, is on a population contrast based on the binary variable sex, though as the authors argue, the same approach might be applicable to other scenarios.

I have only one major and rather general issue and I will phrase my question in a rather non-technical way. Estimating and comparing variances is more difficult than estimating means. (You need larger samples for getting reliable estimates for variances than you need for means.) The explanation that I recall from my stats teacher was: as you have the term (xi-x-bar)^2 in your equation for the variance, the squaring has the effect that it 'magnifies' both sampling variation and measurement error. Hence, the increased variability and the need for larger samples. Of course, the stats teacher also briefly mentioned skew and kurtosis, but with the caveat that with higher moments the uncertainty in your estimates increase (having ^3 and ^4 in the equation). That was the explanation I got, why -even though informative in theory-- people rarely look closely at those higher moments.

Here, the authors do two things: first, they suggest adding a sampling variance for the skewness and kurtosis. Second, in the discussion, they have a section "Limitations", where they recommend a minimum sample size of 50, as skewness and kurtosis estimates can become rather unreliable for smaller samples (tending to produce extreme values).

I think that this sentence is not sufficient. What I would like to see would be a substantially more thorough discussion of how the higher moments are affected by sampling variation and also by measurement error or bias. This might require maybe more that the analysis of one example data set (maybe some simulations or analytic treatment) and also maybe more than a rule-of-thumb threshold number of N=50, which is coming a bit out of the blue. (Not that 50 might not be a sensible number, but I would love to see a bit more, how the authors arrived at this number. Also, I wonder whether the recommendation is actually the same for skewness and kurtosis? If my rather naïve intuition regarding the problem with the higher moments is right, one might need larger samples for kurtosis than for skewness.)

Reviewer #2:

[identifies himself as Andrew W Brown]

The manuscript, "Beyond sex differences in mean: meta-analysis of differences in skewness, kurtosis, and correlation," focuses on estimating and synthesizing comparisons of skewness and kurtosis between sexes in a set of mouse data, as well as comparing correlations. Comparing other aspects of distributions beyond means is important, and there is some work already on handling variance. Skew and kurtosis represent other important aspects of continuous data that are often transformably normal, which may have important considerations in biology, such when traits are under selective pressure in which data in the tails may be more likely to be affected or selected. In this way, the authors bring up an important point. However, the authors discuss these 'effect sizes' without citing or deriving the formulae. As far as I can tell, the approach was to use off-the-shelf calculations of skewness, kurtosis, and correlation and their variances, take a simple difference of the two variables, and assume that the variance of the difference is simply a function of the typical pooled variance. Whereas the central limit theorem works well for means and standard errors, other parameters may not be estimated as well. For example, consider pooling something like maximum lifespan in this way: take the maximum of males and maximum of females and substract. Yet, we know that there are unfavorable statistical properties of sample maxima. It is not obvious to me that shape parameters defining distributions converge like means, nor am I convinced that they are problematic like extrema, but that is for the authors to prove. There are also potential concerns with interpretation in the face of differences in other parameters. Consider interpreting differences in kurtosis or skew in the face of substantial heterogeneity of variance.

Together, these concerns undermine confidence in the meta-analysis of these estimates. The authors meta-analyze these estimated differences by taking the means across sites and strains. We know that means of other distributional parameters (or their differences) do not necessarily behave like means of means (the typical question in meta-analyses). Even means needed bias corrections, with Hedge's g resulting from Cohen's d. There also does not seem to be any estimates of heterogeneity that would be expected from a random effects meta-analysis.

It is also not clear that using a random effects meta-analysis is appropriate for the question, given the authors are comparing different strains and sites. Although the design heterogeneity would necessitate a random effects meta-analysis in other circumstances, it is not clear that there is a coherent distribution of effect sizes that would be appropriately represented by a random effects meta-analysis in this case. For example, the authors have not justified that pooling differences in kurtosis across strains makes scientific sense, when the kurtosis could be different across strains.

Given the authors are interested in these effect sizes in the case of meta-analysis, the authors need to characterize and validate the approach. For instance, the authors could pool all of the data into one 'true' population and empirically set the 'true' parameter values. After establishing the parent distribution, draw multiple samples from the 'true' population, estimate the parameter of interest and its variance, compare to the 'true' parameter, and repeat. Then, get some estimate of performance, like confidence interval coverage. It could be something like this (https://doi.org/10.1038/s41366-020-0554-2), but would need to include some aspect of meta-analysis. If using random effects meta-analysis, the data generating process that gives rise to differences across strains and sites would need to be defined. The authors have the individual-level data, and show site/strain specific distributions, but never define the overall distribution that is being compared; rather, they present only the mean ES.

A couple of other thoughts:

The results discuss individual samples and whether certain estimates cover the null, but the purpose of a meta-analysis is to pool data to answer a coherent question, so emphasizing single studies runs counter to the purpose (e.g., line 187).

The recommendation at 252 to use 50 animals for each group seems completely arbitrary. This needs to be empirically justified (e.g., as the CLT is for sample size of 30).

Figures indicate data are scaled, but no information is given about the scaling.

Reviewer #3:

[IMPORTANT: please see attachment for fully formatted version]

1. Summary of the research and overall impression

This methods-focused article introduces three novel effect size metrics that quantify differences in higher-order moments of data distributions—specifically skewness, kurtosis, and within-group correlation—between two groups, while claiming to account for sampling error. Using a large dataset of mouse traits, the authors demonstrate the potential of these metrics to support more nuanced biological analyses of sex differences, extending beyond the conventional comparison of group means. This application is engaging and relevant, and the manuscript is generally well written and scholarly in tone.

While the underlying statistical concepts are familiar to statisticians and data analysts, the use of standardized effect sizes to quantify differences in skewness, kurtosis, and correlation in a meta-analytic framework is likely to be novel to many readers in the biological sciences. As such, the manuscript may offer value by expanding the toolbox available for quantitative synthesis across studies.

However, I am uncertain whether the proposed metrics address a clearly defined or pressing biological question, as expected by the journal’s scope. Nevertheless, this may be the first attempt to present standardized effect sizes for higher-order distributional properties to a broader audience, along with accessible R code for implementation. I can envision their integration into future meta-analyses involving two-group comparisons.

A major concern relates to the recommended minimum sample size of 50 per group, which is difficult to achieve in many preclinical studies. While I recognize the statistical necessity, this limitation may substantially constrain the applicability of the proposed approach in real-world settings.

Overall, I see potential in the methodological contribution, though I remain uncertain about its future uptake in the biological community.

2. Evidence and examples

Major Issues.

1. Could the authors please clarify what is meant by “accounting for sampling error” as stated in the abstract? In the current version of the manuscript, only the sampling variance of the proposed metrics is reported, which does not by itself constitute a correction for sampling error. If the authors apply any method to reduce or correct for sampling error—such as bias correction, shrinkage estimation, or bootstrapping—this should be clearly described in the methods section. If not, the claim in the abstract should be rephrased to avoid confusion.

2. Could the authors please clarify the origin of formulas (1)–(12)? It is currently unclear whether these equations are derived by the authors, adapted from existing literature, or directly taken from another source. A mathematical reference or derivation would help readers assess the theoretical foundation of the proposed metrics. In the provided R code, reference is made to Pick et al. [18]; if the formulas are based on or adapted from that work, this should be explicitly stated in the manuscript.

3. Would the authors consider citing prior work proposing standardized skewness metrics to enhance comparability? I found for example that Malgady (2007) (How skewed are psychological data? A standardized index of effect size, J Gen Psychol.

2007 Jul;134(3):355–359. doi: 10.3200/GENP.134.3.355-360) introduced a 0–1 scaled index of skewness aimed at improving comparability across studies. Including this reference would acknowledge that using skewness as a comparative metric is not entirely new and place the current work in the context of previous efforts.

Minor Issues

• The phrase “The time is particularly ripe for analyses” is too colloquial and should be rephrased to maintain a formal academic tone.

---

## [Decision Letter · Decision Letter 2]

17 Oct 2025

Dear Dr Pollo,

Thank you for your patience while we considered your revised manuscript "Beyond sex differences in the mean: new approaches to meta-analyse differences in skewness, kurtosis, and correlation" for consideration as a Methods and Resources at PLOS Biology. Your revised study has now been evaluated by the PLOS Biology editors, the Academic Editor and the original reviewers.

You'll see that reviewer #1 now signs off enthusiastically with no further requests. Reviewer #2 complains that you failed to respond to some of his previous comments (you might want to check back and respond to these retrospectively), and while he says that the paper is much improved, he pushes for further analyses and more rigorous reporting of your meta-analyses. Reviewer #3 thinks that their concerns have been minimally addressed (again, you should look back and see if you can address these more fully), but the large sample numbers required mean that his/her scepticism about uptake prevents them from giving a firm recommendation.

In light of the reviews, which you will find at the end of this email, we are pleased to offer you the opportunity to address the remaining points from the reviewers in a revision that we anticipate should not take you very long. We will then assess your revised manuscript and your response to the reviewers' comments with our Academic Editor aiming to avoid further rounds of peer-review, although we might need to consult with the reviewers, depending on the nature of the revisions.

**IMPORTANT - SUBMITTING YOUR REVISION**

*Resubmission Checklist*

*Published Peer Review*

*PLOS Data Policy*

*Blot and Gel Data Policy*

Sincerely,

Roli

Roland Roberts, PhD

Senior Editor

PLOS Biology

rroberts@plos.org

REVIEWERS' COMMENTS:

Reviewer #1:

I am happy with the changes and how the authors answered my question. Congratulations to this nice piece of work.

Reviewer #2:

[identifies himself as Andrew W Brown]

The manuscript, "Beyond sex differences in the mean: new approaches to meta-analyse differences in skewness, kurtosis, and correlation," made many improvements. Although the manuscript seemed to address some of my concerns, oddly there were multiple paragraphs of my remarks that had no responses at all in the response to reviewers.

The manuscript does a much better job assessing the proposed metrics with the simulations. The authors proposed and tested many different situations (1200 scenarios for delta_sk and delta_ku; 768 for delta_Zr), but they do not report whether the metrics are acceptable or particularly problematic, other than based on sample size. Is there any other inference that can be drawn from these different scenarios? As an analogy, consider the different recommendations for assessing differences in central tendency whether data are or are not normally distributed, groups are or are not homoscedastic, samples sizes are or are not equal, etc…

The authors estimate 95% CIs for their metrics in their meta-analysis, but, as I mentioned before, the authors have not demonstrated or established (e.g., through citing other work) that their approach results in appropriate interval coverage. They mention bias and relative bias, which is good to see, but they have not demonstrated or made the case that these extend to what is needed for within-study assessment for any confidence interval that would be included in a meta-analysis.

The meta-analysis itself, then, serves as a demonstration of the use of the metrics in practice, but still does not assess the performance of the individual study metrics in a meta-analytic context. The raw data are there, so comparing the meta-analyzed effect estimate against an individual-rodent-data meta-analysis would inform how well the meta-analyzed metrics match the real data. Additional simulations could inform the performance, as well: How does a meta-analysis perform under the null? How does it perform under different known effect sizes, different variances, and sample sizes? How does it perform under heterogeneity of effects? As I mentioned before, this could be assessed through simulations based on the raw, individual-rodent data by creating subsamples, changing the effect size, variance, and sample size, and introducing heterogeneity of effect sizes.

Even without assessing the performance of these metrics in meta-analysis, the meta-analysis in Figure 3 and the text describing the analyses are missing most of the information that would be expected of a forest plot of a random effects meta-analysis: sample size, effect estimates, heterogeneity estimates and tests, etc… The reporting of the meta-analyses is unsatisfactory by norms and standards (e.g., PRISMA). For instance, in line 259 the authors call out a specific phenotyping centre, but never report whether there was significant heterogeneity to justify evaluating individual effect estimates. In 259-261, the manuscript reports greater skewness in fat mass and heart weight, then follow with the fact that the values overlap zero. They therefore have not specified what their criteria are for concluding differences, but even if they had the meta-analyzed estimates are close to null with wide confidence intervals, making it strange to make conclusions about differences in skewness. The article's title purports to present new approaches, and thus it is incumbent on the authors to demonstrate how to use and rigorously report a meta-analysis of these metrics.

Minor point: Lines 69-71 are presumptive. We do not know why people focus on means, but 'obsession' seems an overstatement, and it is more likely that means are just what people were taught.

Line 245: it is unclear why data were limited. For meta-analysis purposes, it seems it would make sense to include independent samples with varying sizes, which would represent realistic literature.

I invited a junior colleague to provide some thoughts, and he remarked:

They used jackknife estimator to estimate bias under finite sample size and normality assumption is violated, which is more common in real life dataset. And I know it's straight forward to prove that the proposed difference of sample skewness and kurtosis are unbiased under the normality assumption, so perhaps a brief comment on this point would be helpful (I might have missed it if it's mentioned elsewhere).

This paper might be focusing on the practical side of this method (finite sample), but I am interested to see if these estimators are consistent (i.e. as sample size goes to infinity, the difference in sample skewness converge in probability to the difference in population skewness), which might be beyond the scope of this paper.

Reviewer #3:

My review points have all been addressed, though on a very short note within the manuscript (like citing the reference [29] on a standardized skewness metrics, without any elaboration). Overall, I see potential in the methodological contribution, though I remain uncertain about its future uptake in the biological community. Required sample sizes are huge, in the field of preclinical research at least. That is why I stay undecided and can unfortunately neither give a clear recommendation nor recommend rejecting the manuscript.

---

## [Editor Report · Decision Letter 3]

23 Jan 2026

Dear Dr Pollo,

Thank you for your patience while we considered your revised manuscript "Beyond sex differences in the mean: new approaches to meta-analyse differences in skewness, kurtosis, and correlation" for publication as a Methods and Resources at PLOS Biology. This revised version of your manuscript has been evaluated by the PLOS Biology editors and the Academic Editor.

Based on our Academic Editor's assessment of your revision, we are likely to accept this manuscript for publication, provided you satisfactorily address the following data and other policy-related requests.

IMPORTANT - please attend to the following:

a) We need the Title to avoid the colon and to contain an active verb. We suggest: "New approaches to meta-analyse differences in skewness, kurtosis and correlation reveal sex differences beyond the mean" or (presumably it's generalisable beyond sex differences?): "New approaches to meta-analyse differences in skewness, kurtosis and correlation reveal between-group differences beyond the mean"

b) Many thanks for supplying the data and code needed to recreate the Figures in Github. However, because Github depositions can be readily changed or deleted, please make a permanent DOI’d copy (e.g. in Zenodo) and provide this URL (see below).

c) Please cite the location of the data clearly in all relevant main and supplementary Figure legends, e.g. “The data and code needed to generate this Figure can be found in https://zenodo.org/records/XXXXXXXX

d) I note that you mention me and the reviewers (“Andrew W. Brown, two anonymous referees, and the editor Roland Roberts”) in the Acknowledgements. While we appreciate the sentiment, this is against PLOS policy, so please could you remove this?

e) Please include the URLs of your funders in the Financial Disclosure statement.

We expect to receive your revised manuscript within two weeks.

*Published Peer Review History*

*Press*

Sincerely,

Roli Roberts

Roland Roberts, PhD

Senior Editor

rroberts@plos.org

PLOS Biology

CODE POLICY

Per journal policy, if you have generated any custom code during the course of this investigation, please make it available without restrictions. Please ensure that the code is sufficiently well documented and reusable, and that your Data Statement in the Editorial Manager submission system accurately describes where your code can be found. More information on our Code Policy, what and how to share can be found here: https://journals.plos.org/plosbiology/s/code-availability

DATA NOT SHOWN?

---

## [Editor Report · Decision Letter 4]

29 Jan 2026

Dear Dr Pollo,

Thank you for the submission of your revised Methods and Resources paper "New approaches to meta-analyse differences in skewness, kurtosis, and correlation" for publication in PLOS Biology. On behalf of my colleagues and the Academic Editor, Marcus Munafò, I'm pleased to say that we can in principle accept your manuscript for publication, provided you address any remaining formatting and reporting issues. These will be detailed in an email you should receive within 2-3 business days from our colleagues in the journal operations team; no action is required from you until then. Please note that we will not be able to formally accept your manuscript and schedule it for publication until you have completed any requested changes.

Sincerely,

Roli Roberts

Senior Editor

PLOS Biology

rroberts@plos.org